# Characterization of Indoor Molds after Ajka Red Mud Spill, Hungary

**DOI:** 10.3390/pathogens13010022

**Published:** 2023-12-26

**Authors:** Donát Magyar, Zsófia Tischner, Bence Szabó, Ágnes Freiler-Nagy, Tamás Papp, Henrietta Allaga, László Kredics

**Affiliations:** 1National Center for Public Health and Pharmacy, H-1097 Budapest, Hungary; 2Department of Environmental Safety, Institute of Aquaculture and Environmental Safety, Hungarian University of Agriculture and Life Sciences, H-2100 Gödöllő, Hungary; zsofi.tischner@gmail.com; 3Centre for Translational Medicine, Semmelweis University, H-1085 Budapest, Hungary; bencetra@gmail.com; 4Department of Animal Hygiene, Herd Health and Mobile Clinic, University of Veterinary Medicine, H-1078 Budapest, Hungary; freiler-nagy.agnes@univet.hu; 5HUN-REN-SZTE Pathomechanisms of Fungal Infections Research Group, University of Szeged, H-6726 Szeged, Hungary; pappt@bio.u-szeged.hu; 6Department of Microbiology, Faculty of Science and Informatics, University of Szeged, H-6726 Szeged, Hungary; henrietta.allaga@gmail.com (H.A.); kredics@bio.u-szeged.hu (L.K.)

**Keywords:** red mud, bauxite processing residue, indoor mold, fungal growth

## Abstract

A red mud suspension of ~700,000 m^3^ was accidentally released from the alumina plant in Ajka, Hungary, on the 4th of October 2010, flooding several buildings in the nearby towns. As there is no information in the literature on the effects of red mud on indoor mold growth, we conducted studies to answer the following question: does the heavy metal content of red mud inhibit fungal colonization in flooded houses? In order to gain knowledge on fungal spectra colonizing surfaces soaked with red mud and on the ability of fungi to grow on them, swabs, tape lifts, and air samples were collected from three case study buildings. A total of 43 fungal taxa were detected. The dominant species were *Penicillium* spp. on plaster/brick walls, but *Aspergillus* series *Versicolores*, *Cladosporium*, *Acremonium*, and *Scopulariopsis* spp. were also present. The level of airborne penicillia was high in all indoor samples. Selected fungal strains were subcultured on 2% MEA with 10^−1^ and 10^−4^ dilutions of red mud. The growth rate of most of the strains was not significantly reduced by red mud on the artificial media. The consequences of similar industrial flooding on indoor molds are also discussed in this paper.

## 1. Introduction

The red mud is a solid waste product of bauxite processing via the Bayer process, which refines bauxite (aluminium ore) into a form of aluminum oxide called alumina with sodium hydroxide [1]. Red mud is a caustic (pH > 12.8–13.0) material consisting of Fe_2_O_3_ (40–45%), Al_2_O_3_ (10–15%), SiO_2_ (10–15%), CaO (6–10%), TiO_2_ (4–5%), and Na_2_O (5–6%) in hydrated forms [2]. Minor or trace elements in this material can be K, Cr, As, Hg, V, Ni, Ba, Cu, Mn, Pb, Zn, Zr, Y, Sc, Ga, etc., and even ^232^Th and ^238^U, which have radioactivity about 10-fold above the soil background [3,4,5,6,7,8]. On the 4th of October 2010 the dam of red mud reservoir number X (ten) of the Ajkai Timföldgyár Zrt. alumina plant collapsed, freeing approximately 700,000 m^3^ of liquid waste from red mud lakes. Red mud suspension was released as a 1–2 m wave, affecting about 40 km^2^ of land, the Torna-Marcal river system, and flooding several buildings in the nearby localities, including the village of Kolontár and the town of Devecser, leading to 10 human deaths and extensive property damage. On the 22 November 2010, indoor mold growth was reported by the local agencies.

A review of large literature databases concluded that higher respiratory morbidity and allergic complaints have been observed in occupants of mold-colonized structures [9]. The most important factors influencing fungal growth are moisture, type of substrate, temperature, and exposure time [10]. In relation to moisture, water activity (a_w_, the amount of available/unbound water in the substrate) is one of the most important factors for indoor mold growth. Indoor fungi vary in their a_w_ requirements, and this ranges from 0.69 to >0.94 [11]. High (>0.90) water activity can be found in building materials if they have been affected by flooding [12]. Mold problems caused by extensive floodings were studied by different authors [9,13,14,15,16,17,18], especially after the hurricanes Katrina, Rita, and Maria [19,20,21,22,23]. These floodings, however, did not contain chemical pollutants at such high concentrations [24] as industrial sludges like red mud. The disaster had fatal consequences for living organisms in soils and rivers [25]. However, the effect of red mud on the indoor fungal spectra seems to be completely unknown. Reviewing the literature on indoor mycology, no similar catastrophes were documented. As about 90 million tons of this waste material are produced globally [26], the risk of further red mud outflows cannot be excluded. One may hypothesize that red mud has a negative impact on indoor fungi, considering its pH and heavy metal content; however, the complexity of the environmental factors—such as substrates and exposure time—makes the outcomes hardly predictable. As the fungal isolates from the Hungarian red mud catastrophe were available in a microbial collection, we aimed to study the effects of red mud on the composition and growth of indoor molds.

## 2. Materials and Methods

### 2.1. Microbiological Sampling and Identification

Swab, tape-lift, and air samples were collected on the 21 March 2011 in three case study buildings in Devecser affected by the red mud catastrophe. The first building represented the most common type of family house in Devecser, with plaster/brick walls and tile flooring. The second building was a wooden shed, often used as an annex in Hungary. The third building was a root cellar with brick walls and cob flooring—this type is often used as a wine cellar; the basement of the family house, or as an outbuilding (Figure 1).

Air sampling was carried out with the SAS IAQ^TM^ device. A total of 100 L of air were collected from indoor and outdoor places onto 2% Malt Extract Agar (MEA) and Czapek-Yeast Extract Agar media supplemented with 0.1 g L^−1^ chloramphenicol. The samples were incubated at 25 °C for 5 days. After incubation, the colony-forming units (CFUs) were counted, and the concentrations (CFU m^−3^) were calculated using the ‘positive hole correction’ method [27,28]. The isolated fungi were subcultured on MEA and identified morphologically [11] by a light microscope (Zeiss Jenaval, Germany) at 600× magnification, then deposited and maintained in the Szeged Microbiology Collection (SZMC) at the University of Szeged, Hungary.

For the purposes of molecular identification, the mycelia of the fungal isolates grown in liquid YEG medium (5 g L^−1^ glucose, 1 g L^−1^ yeast extract, and 5 g L^−1^ KH_2_PO_4_ in distilled water) were ground into dust with a mortar and pestle in liquid nitrogen. The extraction of total genomic DNA was carried out with the GenElute^TM^ Plant Genomic DNA Miniprep Kit (Sigma-Aldrich) according to the protocol provided by the manufacturer. The internal transcribed spacer (ITS) region of the rRNA gene complex, incorporating ITS 1, the 5.8S rRNA gene, and ITS 2, was amplified using primers ITS1 (5′-TCCGTAGGTGAACCTGCGG-3′) and ITS4 (5′-TCCTCCGCTTATTGATATGC-3′) [29]. All amplifications were performed in a MJ Mini^TM^ Personal Thermal Cycler (Bio-Rad Laboratories) with the following temperature profile: 5 min initial denaturation at 94 °C, followed by 35 cycles of 30 s denaturation at 94 °C, 40 s primer annealing at 48 °C, 1 min extension at 72 °C, and a final extension step of 3 min at 72 °C. PCR products were subjected to purification and automatic sequencing with primer ITS4 at Beckman Coulter Genomics, Essex, England. The resulting sequences were deposited in the GenBank database of the National Center for Biotechnology Information (NCBI) under accession numbers OR676943-OR676961. Sequence analysis was carried out by a BLASTN similarity search [30] on the website of NCBI http://www.ncbi.nlm.nih.gov/blast/ (accessed on 31 October 2023).

### 2.2. Colony Growth Tests

As a complex effect of the red mud on fungi can be assumed due to its heavy metal content, pH, and water activity, we studied these 3 factors separately. The effects of red mud, pH, and water activity on the growth of fungal isolates (N = 15) were tested on different agar media.

#### 2.2.1. Red Mud Tests

To test the effect of red mud on the fungal isolates, a total of 500 g of red mud was collected near the case study buildings in Devecser in March 2011 in order to create special agar media. The chemical content of the red mud sample was analyzed. Trace analysis quality reagents were used for sample preparations: hydrochloric acid (37 m/m%, Aristar) and nitric acid (69 m/m%, Aristar). Mono- and multielement standards purchased from Perkin Elmer Inc. (Shelton, CT, USA) and VWR International Ltd. (Leicestershire, UK) were used for the calibration of the inductively coupled plasma optical emission spectrometer (ICP-OES). To clean the laboratory equipment, 0.15 M hydrochloric acid solution (37 m/m%, Aristar) was used. All the chemicals were obtained from VWR International Ltd. (Leicestershire, UK). Solutions were appropriately diluted by deionized water, which was produced by a Purite Select Fusion 160 BP water purification system (Suez Water Ltd., Thame, UK). The elements were determined by the ICP-OES Avio 550 Max system (Perkin Elmer, Shelton, USA), which works with the following parameters: RF generator at 40 MHz solid state, flat plate plasma technology, free running; RF power at 1300 W; plasma gas flow rate at 12 dm^3^min^−1^; auxiliary gas flow rate at 0.2 dm^3^min^−1^; nebulizer gas flow rate at 0.7 dm^3^min^−1^; nebulizer type Burgener Peek Mira Mist; observation height at 15 mm. Argon gas of 4.8 purity was used for ICP measurements (Messer Hungarogáz Ltd., Budapest, Hungary). The so-called total elemental concentration can be determined by the method of leaching with *aqua regia* [5,31]. For sample preparation, 0.25 g of sample was weighted into a glass baker, and 3.8 mL of hydrochloric acid and 1.2 mL of nitric acid were added. The solution was evaporated up to almost dry on a steam bath, then the residue was dissolved in 3 mL of 1:1 hydrochloric acid. The solution was filled up to 50 mL in a volumetric flask. The red mud sample was prepared in three parallels. After leaching, the samples were diluted by deionized water and measured by ICP-OES with the addition of 1 mg L^−1^ yttrium (Y) solution as the initial standard and 0.25 mg L^−1^ gold (Au) solution for the stabilization of mercury content. The detection limits were calculated as three times the standard deviation of “blank” concentrations.

Red mud was added to MEA in different concentrations at 10^−1^, 10^−2^, 10^−3^, and 10^−4^ m/m%, having pHs of 7.0, 6.8, 5.2, and 5.2, respectively. Based on the previous growth tests, MEA media with 10^−1^ and 10^−4^ red mud concentrations were selected for further studies. To determine the water activity value of the red mud media, a LabTouch-aw instrument (Novasina AG, Lachen, Switzerland) was used.

#### 2.2.2. pH Tests

The effect of pH was tested on salty agar media. Seven pH values (2.2, 3.0, 4.0, 5.0, 6.0, 7.0, and 8.0) were selected to cover the above-mentioned range using McIlvain buffer solutions (mixtures of 0.3 M Na_2_HPO_4_ × H_2_O and 0.1 M citric acid in different proportions) in double-concentrated MEA.

#### 2.2.3. Water Activity Tests

The effect of water activity (a_w_) was tested by using NaCl in media containing yeast extract agar (YEA, 10 g L^−1^ glucose, 5 g L^−1^ KH_2_PO_4_, 1 g L^−1^ NaNO_3_, 2 g L^−1^ yeast extract, 1 g L^−1^ MgSO_4_ × 7H_2_O, 15 g L^−1^ agar in distilled water), according to the data published earlier [32]. A_w_ was calculated from the osmolality values as described earlier [33]. Eight concentrations of NaCl were tested: 0% NaCl (a_w_ 0.997), 1% NaCl (a_w_ 0.991), 3% NaCl (a_w_ 0.980), 5% NaCl (a_w_ 0.968), 6% NaCl (a_w_ 0.962), 8% NaCl (a_w_ 0.951), 9% NaCl (a_w_ 0.945), and 12% NaCl (a_w_ 0.922).

#### 2.2.4. Measurements and Statistics

For the determination of colony growth rates, 15 fungal strains isolated from water-damaged buildings were selected: eight strains—TD4-TD6; TD8; TD12; TD16; TD22; and TD25—were isolated from buildings flooded by red mud; strain TD9 was collected from the outdoor air in the town after the red mud disaster, while six strains were collected from water-damaged buildings in Hungary; and the curtain of a Hungarian hostel room was used as a control (Table 1). Fungal strains were inoculated centrally into 90 mm Petri plates with 4 mm-diameter plugs cut from the margin of actively growing colonies, then incubated at 25 °C for 5 days. Colony diameters (mm) were measured as a response variable. As fungal metabolism may alter the acidity of the substrate, the pH of the medium was also measured daily. The tests were performed in three replicates. The growth of each replicate of each strain was characterized by their cumulative colony diameter (the sum of the colony diameters on each of the 5 days). The impact of treatment on the colony diameter of fungi was analyzed using the Analysis of Variance (ANOVA) method, with a preset alpha value of 0.05. This robust statistical tool allowed for the assessment of differences among multiple treatment groups, providing valuable insights into the overall treatment effect. To further elucidate the specific variations between individual treatments, a Dunnett’s post-hoc test was subsequently conducted with the measurement data of control treatments (a_w_ control = 0.0% NaCl, pH control = MEA, red mud 10^−1^ control = red mud 10^−4^), to which all comparisons were made. This post-hoc analysis enabled a detailed comparison of each treatment group against a control group, pinpointing any treatments exhibiting statistically significant differences in colony diameter. In the case of the red mud agar treatment, where only two levels of treatment were compared, the results of the ANOVA and Dunnett’s test are equivalent, as only one pairwise comparison of two groups was necessary.

## 3. Results

A total of 43 fungal taxa were detected in the buildings inundated by red mud; 26 taxa were isolated from indoor air, while 21 taxa were collected from surfaces covered by industrial waste residue (Table 2). Intensive mold growth was observed on the wet plaster wall of the living room on the first floor, flooded at a height of 1.5 m, where the dominant taxa were *Penicillium* section *Chrysogena*, but *Cladosporium*, *Acremonium*, and *Scopulariopsis* spp. were also detected. Different compositions of fungi were observed in the shed, where wet wood (walls, cupboards, and firewood) was colonized by a more variable mycota (*Acremonium*, *Alternaria*, *Aspergillus*, *Cephalotrichum*, *Chaetomium*, *Cladosporium*, *Penicillium*, *Scopulariopsis*, *Sepedonium*, and *Ulocladium* spp.). A high number of spores and red mud particles were observed in Psocoptera spp. fecal pellets in surface samples.

The airborne concentration of fungi is given in Table 3. The level of airborne penicillia was high in all samples, especially in the brick building, where the concentration was 1700-fold higher than outdoors. *Penicillium* sp., section *Chrysogena*, *Penicillium* sp., section *Penicillium*, and *Penicillium* sp., section *Fasciculata*, and series *Camemberti* were the dominant species in the air of the brick building, wooden shed, and cellar, respectively. *Cladosporium* spp. and *Parengyodontium album* also had remarkable concentrations in the cellar.

The radial growth test showed statistically significant differences in colony diameters under the different treatment regimes (Table 4, Figure 2, Appendix A). Salt concentrations (water activity) and pH had a significant effect on all strains based on the results of the ANOVA-s (Table 4). Based on the pairwise comparisons to the control treatments conducted with Dunnett’s tests, two strains (TD8, T401B) tolerated low water activity (a_w_ 0.922), but most of the tested fungi showed poor development on it. Some strains (TD4, TD6, TD9, TD12, TD22, TD25) did not tolerate even moderate a_w_ values (0.954–0.962). High a_w_ was preferred by TD4, TD5, TD6, TD9, TD12, TD22, TD25, and T399 (Figure 2, see Appendix A for detailed results of the Dunnett’s test).

Most strains preferred the pH range of 5.0–7.0. Low pH was not tolerated by TD6 (pH ≤ 4.0), as well as T398B, TD8, and TD25 (pH = 2.2), while the other stains tolerated it poorly, showing delayed and reduced growth. Growth of TD4 was inhibited by pH 8 and 7, while this fungus preferred pH 3. TD5 grew well on all pH values (Figure 2, see Appendix A for detailed results of the Dunnett’s test).

The resulting chemical composition of the red mud sample is given in Table 5. The main chemical elements of the red mud sample were Fe, Ca, Al, and Na. Converted to oxide, we obtained the following concentrations for the main elements: 37% Fe_2_O_3_, 12% Al_2_O_3_, 7.6% CaO, and 3.8% Na_2_O. Elements such as K, Mn, S, P, Cr, Sr, Ni, Pb, Zn, Cu, Co, and Cd were found in minor or trace amounts. The concentrations of Mo, Se, and Hg were under their respective detection limits.

When supplemented with MEA in 10^−1^ concentration, red mud did not lower the water activity (a_w_ 0.97 ± 0.02, i.e., no drought stress during in vitro conditions). The pH of the red mud media did not change markedly during the incubation period (pH 6.50 ± 0.30 on the 1st day and pH 6.65 ± 0.15 on the 5th day in the 10^−1^ dilution of red mud).

All tested strains were able to grow on red mud agar, including controls. High red mud concentrations (10^−1^ dilution of the original sample) significantly affected the growth of 8 strains (Table 4). The combined effect of different culture media to test red mud, its pH, and a_w_ on fungal growth is shown in Figure 2. A weak positive effect of red mud was observed in one case (TD25). A negative effect of red mud was found in seven strains. Delayed initial growth was detected in 3 control strains (T399, SZMC 2725, and T398A). A smaller colony diameter was measured at the end of the incubation period in TD8, TD9, and TD16 and in the control strain T398B. However, in two of them, the negative effect was possibly due to the pH of the red mud media but not its heavy metal content (see Figure 2, TD9, and TD16). High red mud concentrations were tolerated by the remaining strains without any reduction in growth. Based on the growth profile of the individual strains, the 10^−4^ dilution of red mud had no effect on any strains (Figure 2, see Appendix A for detailed results).

## 4. Discussion

As a result of the tragic industrial accident, aluminum industrial waste residue sludge inundated settlements. The damage first arose from the highly alkaline pH of the red mud, causing severe chemical burns on humans and animals and extirpating the fauna in the contaminated soils and Torna-Marcal river systems [25]. Fungi were probably no exception, and the high pH could also have had a negative effect on them. Most fungal species are considered to germinate and grow well in the weakly acidic to neutral pH range [34]. When spilled, caustic red mud wiped out all detectable forms of life in the Torna river, including aquatic fungi, mainly due to its high pH level (12.8) [35]. However, results suggest that fungi have high colonization capacity or show high resistance to environmental disasters that are as destructive as the red-sludge catastrophe [35]. Although we do not have any information on indoor fungi in the buildings before the disaster, it can be assumed that the indoor hyphomycetes (if any) were killed by the highly alkaline red mud. The saturated building materials left behind by the flood offered a suitable habitat for fungal growth after the dissipation of the acute effects of high pH, possibly by dilution and a progressive carbonation of the sodium hydroxide by atmospheric CO_2_. Laboratory tests showed rapid carbonation and decreasing pH in an aerated and stirred red mud suspension, reaching pH 7.0 in 3 days (Dr. Zoltán Szabó, personal communication). The time required for neutralization possibly depends on the accessibility of CO_2_ as well as the porosity and thickness of the red mud. Reports on the delayed mold growth support the hypothesis of a progressive decrease in the initial pH. It cannot be excluded that the pH of the red mud was lowered by fungal activity. Generally, the CO_2_ produced by the respiration of fungi during metabolism forms carbonic acid when it reacts with water molecules. Laboratory experiments show that *Aspergillus niger* can decrease the pH of the substrate by excreting acidic metabolites (citric, gluconic, and oxalic acid, the first being the dominant) [36]. By monitoring the changes in the pH values of red mud-based media, *Penicillium tricolor* RM−10 was found to drastically reduce the pH value of the medium from over 10.0 to 3.0 in 200 h [37].

The chemical composition of the red mud from Ajka was measured by several studies. The results of the sample in the present study are in good agreement with previous findings [5,6]. The toxicity of red mud was related to the release of oxyanion-forming metals and metalloids. The trace metal content of red mud depends on the original source of the bauxitic ore from which the alumina was refined [38]. Governmental agencies stated that the heavy metal concentrations were not considered dangerous for the environment [39]. In red mud samples collected after the catastrophe in Hungary, a relatively low concentration of heavy metals was measured, although still about seven times the levels in normal soil [39]. According to our growth tests, most (10 of 15) fungal strains isolated from building materials covered with red mud tolerated its heavy metal content and grew well in the typical range of pH and water activity characterizing the red mud substrate. Isolates tolerating red mud poorly belonged to *Aspergillus* sp., series *Versicolores* (T398B), *Penicillium expansum* (SZMC 2725), *Penicillium* section *Chrysogena* (T08), and *Penicillium* spp. (T398A, T399). Interestingly, isolates well tolerating red mud were their close relatives, e.g., TD12 (*Aspergillus* sp., series *Versicolores*) and TD401A (*P*. *chrysogenum*). This suggests that red mud tolerance may be determined by infraspecific genetic factors.

At the time of the spillout, the original pH of the red mud was highly alkaline (pH > 12.8–13.0) [2]. Once the near-neutral pH was reached, the red mud did not inhibit fungal colonization (in some strains, delayed colonization was observed). The growth rate of most of the strains (10/15) was not significantly reduced by red mud on the artificial media, indicating no difference in mold growth in the buildings flooded with or without red mud. The positive effect of red mud was not observed either. Fungi formerly known from the red mud ponds of alumina factories (e.g., *Penicillium tricolor*, *Trichoderma asperellum*) [37,40] were not detected in the flooded buildings in Devecser. Control strains (isolated from water-damaged buildings without red mud) also tolerated red mud in our colony growth tests. This suggests that the chemical composition of red mud had a nil or low selective effect on fungi. Predominantly *Penicillium* section *Chrysogena* and *Aspergillus* series *Versicolores* were detected in the buildings flooded by red mud, which are frequent fungi in water-damaged buildings in Hungary (Donát Magyar, unpublished). Based on the analysis of 28 mold-infested buildings, a trend of higher prevalence of indoor *Aspergillus* and *Penicillium* spp. was observed in buildings with previous flooding histories. In the indoor environment, flooding history was found to be the strongest determinant of the prevalence of the above-mentioned genera [16]. In line with our results, Baldrian [41] reports that higher rates of spore germination of *Penicillium chrysogenum* (*P.* section *Chrysogena*) were associated with a high level of water activity, but pH did not have a significant effect [42]. Urík et al. [43] reported fungal growth on red mud, where fungi formed well-distinguished aerial mycelia, except *P. chrysogenum*, which grew in the form of submerged mycelium. In this experiment, the presence of red mud in the culture medium resulted in lesser sexual and asexual *Aspergillus* biomass production compared to *Penicillium* species. Extremely high resistance of *Penicillium* species has been recognized on highly heavy metal-contaminated substrates [44]. Qu et al. [36] carried out growth tests at a high (5%) pulp density of red mud. The experiment demonstrated reduced fungal growth, possibly because red mud particles covered almost the whole mycelial surface, which possibly formed a thick barrier and concentrated toxic metal ions around it. Such large deposits of red mud were found sparsely in our buildings (behind door casings and the decorative panel of the bathtub), but more frequently, they were present as a thin layer on the surfaces. Apparently, the main environmental factor affecting fungal colonization was the high water content of the building materials. Homes with more than ~1 m of indoor flooding demonstrated higher levels of mold growth compared with homes with little or no flooding [22]. According to other studies, it is important how long flooding water stays in the building [45]. In the current case, however, the red mud tsunami was shortly drained by the rivers. On the other hand, attempts to remove red mud from some houses using high-pressure clean water probably caused more water to enter the buildings’ envelope. The timing of the survey also has an important effect. Sampling a few days after the flooding showed the presence of *Aspergillus*, *Cladosporium*, and *Penicillium* spp. in the dwellings [19,46], whereas a survey performed 4 months after the flooding resulted in *Alternaria* and *Stachybotrys* spp. [45]. In our case, both groups of fungi were present, and their occurrence depended on the type of substrate. Brick/plaster walls were dominated by *Penicillium*, while wooden materials showed a higher diversity of fungi. The higher fungal diversity on indoor wooden materials than on plaster is a common phenomenon [47]. Fungal species present on the walls flooded by red mud belonged to common molds frequently found on water-damaged buildings in Hungary [48]. Apparently, after the red mud disaster, mold colonization of houses was similar to any natural flood.

Post-flood fungal colonization of indoor environments represents a health threat to inhabitants, predominantly due to the inhalation of allergens. Several studies have explored the levels of airborne fungi in water-damaged indoor environments and their role in chronic respiratory diseases like rhinosinusitis and asthma. Measurement of airborne viable fungal concentration is informative to assess indoor air quality in flooded buildings [49], although the observation of visible mold is sufficient to inform the public about the health risk and to take preventive measures [50]. Effective health communication in floods is very important [51,52,53]. Post-flood surveys could help to develop evidence-based public health messages to advise homeowners and flood-affected communities on recovery after such disasters. Hoppe et al. [53] showed that proper post-flood remediation led to improved air quality and lower exposures among residents living in flooded homes. Drying water-damaged material thoroughly and fast enough to prevent mold growth is difficult after large-scale water excursions associated with floods [9], therefore the application of high-performance dehumidifiers is recommended. The use of protective personnel equipment and exposure mitigation techniques among residents in order to prevent respiratory morbidity should be encouraged.

It is worth mentioning that floodings represent multiple health risks (statical damage, waterborne diseases, etc.), of which the above-mentioned negative respiratory effect of molds is only one [54]. As the flooded buildings were considered dangerous after the red mud catastrophe, they were evacuated. Subsequently, it was decided to demolish 225 out of 272 and 35 out of 53 damaged houses in Devecser and Kolontár, respectively [55]. This action hindered further research, but a review of the literature allowed us to hypothesize an additional potential health risk. A toxicological hazard may be posed by acidic (pH 3–4) metabolites of fungi, which can metabolize toxic ions from indoor red mud, including arsenic. An example is the well-known public health scandal of the 1900′s, when arsenic was volatilized as trimethylarsine from pigmented wallpapers due to the metabolic activity of *Scopulariopsis brevicaulis* [56,57]. The combination of arsenic and *Scopulariopsis* sp. was also found after the red mud catastrophe; thus, a toxicological risk was theoretically possible and may be considered in a similar disaster with heavy metal exposure.

## 5. Conclusions

Indoor fungi detected after the red mud catastrophe belonged to species frequently reported from naturally flooded buildings. Our results indicated that only a small number of fungal strains were significantly affected by the red mud on the artificial media, indicating only a slight difference in mold colonization in the buildings flooded with or without red mud. The heavy metal content of red mud had a negligible role in the inhibition of fungal growth or composition, but other aspects, e.g., modification of metabolite production, are still largely unknown. Among the many negative outcomes of flooding combined with chemical pollution, the effects of fungal colonization may be complex and hardly predictable; therefore, extra caution should be used.

## Figures and Tables

**Figure 1 pathogens-13-00022-f001:**
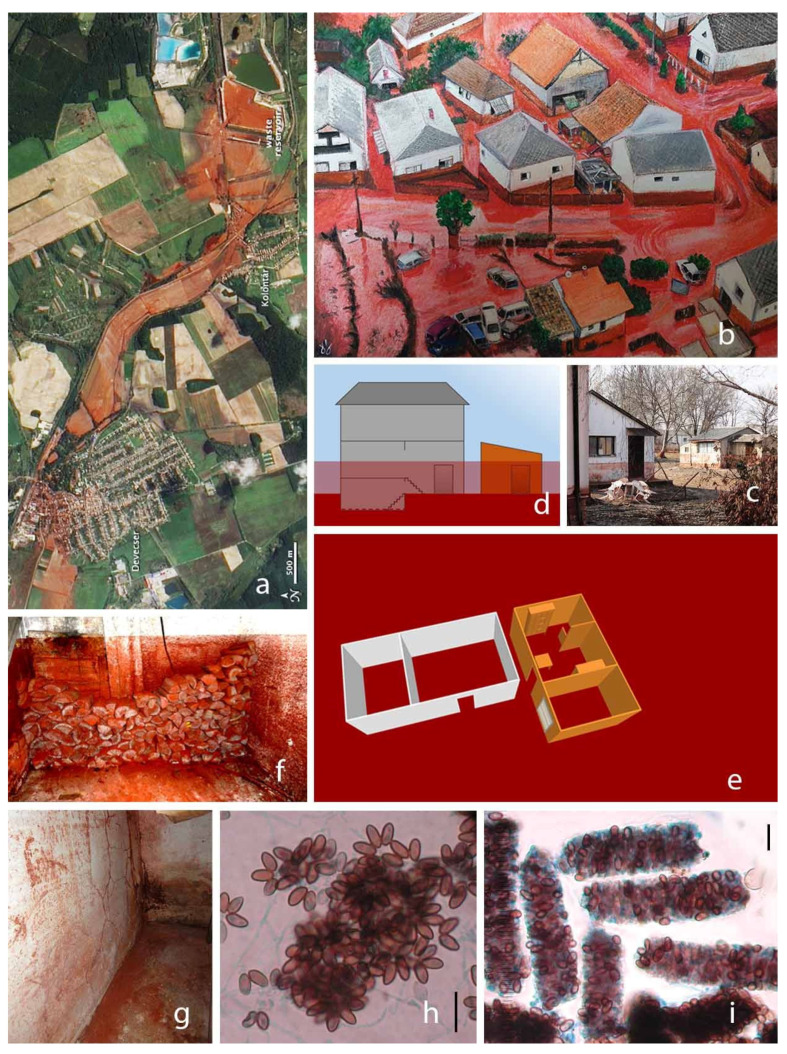
(**a**) A natural-color image of the area surrounding the red mud spill in Hungary. The alumina plant appears along the right edge of the image, with blue and red reservoirs. The red mud forms a red streak running west from the alumina plant. Photo: NASA Earth Observatory. (**b**) Streets of Devecser after the red mud disaster (aquarelle painting of the author (D. Magyar) based on the aerial photo of Sándor H. Szabó. The artwork was painted with red mud collected in the town after the catastrophe. (**c**) Flooded buildings (Photo: D. Magyar). (**d**) Cross-section and (**e**) floorplan of the investigated buildings. (**f**) Firewood covered by red mud. (**g**) Plaster walls covered by red mud in the investigated buildings. (**h**) *Wardomyces inflatus* colony in a surface sample. (**i**) Spores and red mud particles in Psocoptera pellets. Bar = 10 µm.

**Figure 2 pathogens-13-00022-f002:**
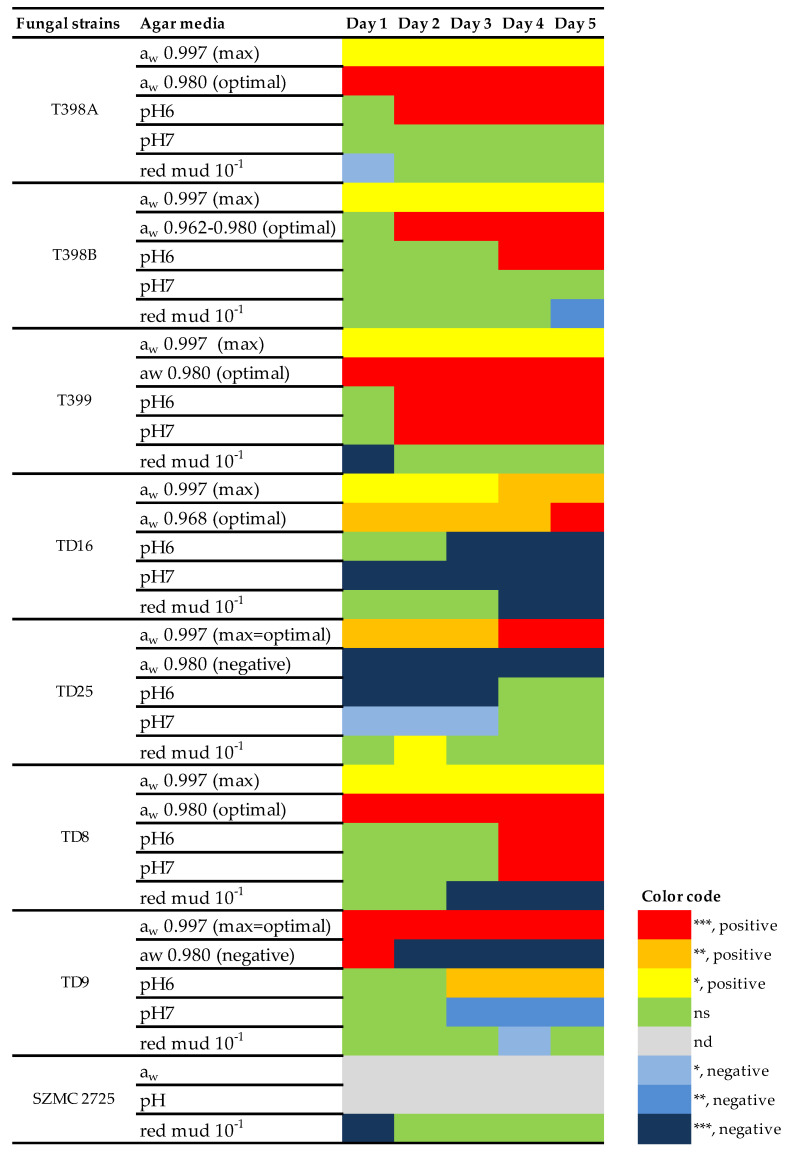
The effect of red mud, pH, and water activity (a_w_) on the colony diameters of fungal strains on special media (MEA + 10^− 1^ red mud concentration, MEA + McIlvain buffer solutions, and YEA + NaCl, respectively) when compared to their growth on the control media (red mud 10^−1^ control = MEA + red mud 10^−4^, pH control = MEA, a_w_ control = YEA). Ranges of a_w_ (0.97 ± 0.02) and pH (6.50 ± 0.30) corresponding to MEA with 10^−1^ red mud concentration are selected. Strains where red mud media had no significant effect are excluded. Significant positive effects are marked with red, orange, and yellow; negative effects are marked with blue. ns = non-significant; nd = no data (not studied). Significant results are marked with ***: *p* < 0.001, **: *p* < 0.01, and *: *p* < 0.05.

**Table 1 pathogens-13-00022-t001:** Fungal isolates used as controls in growth tests.

Taxa	Location	Building Type	Room	Sample	Code	SZMC Number
*Penicillium* sp.	Budapest	family house	bathroom	mold on wall (swab)	T398A	SZMC 22652
*Aspergillus* series *Versicolores*	Budapest	family house	childrens’ room	air	T398B	SZMC 22653
*Penicillium* sp.	Budapest	family house	bedroom	saltpeter under wallpaper (swab)	T399	SZMC 22654
*Penicillium chrysogenum*	Szentendre	family house	cellar	air	T401A	SZMC 22656
*Penicillium* sp.	Szentendre	family house	cellar	air	T401B	SZMC 22657
*Penicillium expansum*	Szeged	college	bedroom	curtain	-	SZMC 2725

SZMC: Szeged Microbiology Collection.

**Table 2 pathogens-13-00022-t002:** Fungal taxa detected in buildings inundated by red mud and outdoors.

Fungal Taxon	Sample Type	Sampling Point	Strain Number	SZMC Number	GenBank Accession Number (ITS)
	**T** = tapelift **S** = swab **A** = air	**b** = brick building **w** = wooden shed **c** = cellar **o** = outdoor			
Ascomycota					
*Acremonium* sp. 1–2	TA	wo			
*Alternaria* sp.	T	w			
*Apiospora arundinis*	S	b	TD19	SZMC 12680	OR676943
*Aspergillus* sp., section *Nidulantes*	A	c	TD4	SZMC 12665	OR676944
*Aspergillus* sp., series *Versicolores*	A	c	TD2b	SZMC 12663	OR676945
*Aspergillus* sp., series *Versicolores*	A	c	TD12	SZMC 12673	OR676947
*Aspergillus* sp., series *Versicolores*	A	w			
*Aspergillus* sp., series *Versicolores*	A	o	TD9	SZMC 12670	OR676946
*Beauveria* sp.	A	c	TD5	SZMC 12666	OR676948
*Beauveria* sp.	A	o			
*Botrytis* sp.	A	w			
*Cephalotrichum* sp.	AT	cw	TD6, -	SZMC 12667	
*Chaetomium* sp.	T	w			
*Cladosporium cladosporioides*-type	AAA	cwo			
*Cladosporium herbarum*-type	AA	co			
*Cladosporium* spp.	TA	bw			
*Geomyces* sp.	T	w			
*Parengyodontium album*	A	c	TD2a	SZMC 12662	OR676953
*Penicillium brevicompactum*	S	b	TD14	SZMC 12675	OR676954
*Penicillium buchwaldii*	S	w	TD21	SZMC 12682	OR676955
*Penicillium buchwaldii*	S	w	TD22	SZMC 12683	OR676956
*Penicillium* sp., section *Chrysogena*	A	b	TD8	SZMC 12669	OR676958
*Penicillium* sp., section *Chrysogena*	S	b	TD13	SZMC 12674	OR676959
*Penicillium* sp., section *Chrysogena*	S	w	TD15	SZMC 12676	OR676960
*Penicillium* sp., section *Chrysogena*	A	o			
*Penicillium* sp., section *Fasciculata*, series *Camemberti*	A	c	TD1	SZMC 12661	OR676957
*Penicillium* sp., section *Penicillium*	A	w	TD16	SZMC 12677	OR676961
*Penicillium* sp. 1	A	o			
*Scopulariopsis* sp.	AT	c	TD7		
*Sepedonium* sp.	T	w			
*Ulocladium* sp.	T	w			
*Wardomyces inflatus*	T	w			
yeast sp. 1	A	c			
yeast sp. 2	A	o			
unknown sp. 1	T	w			
unknown sp. 2	A	c	TD3a	SZMC 12664	
unknown sp. 3/*Penicillium freii*?	A	o	TD11	SZMC 12672	
unknown sp. 4	T	w			
*Alternaria* sp./unknown sp. 5	S	w	TD17		
unknown sp. 6	S	b	TD20	SZMC 12681	
Basidiomycota					
*Bjerkandera adusta*	A	o	TD10	SZMC 12671	OR676949
Mucoromycota					
*Mucor circinelloides*	S	w	TD23	SZMC 12684	OR676950
*Mucor circinelloides*	S	w	TD25	SZMC 12686	OR676951
*Mucor hiemalis*	S	w	TD18	SZMC 12679	OR676952
*Mucor plumbeus*	A	c			
*Mucor* sp.	S	c	TD24	SZMC 12685	
*Rhizopus stolonifer*	A	w			

SZMC: Szeged Microbiology Collection. ITS: internal transcribed spacer.

**Table 3 pathogens-13-00022-t003:** Airborne concentration of the fungal taxa (avg. CFU m^−3^).

Airborne Fungal Taxa	Outdoor	Brick Building	Cellar	Wooden Shed
*Acremonium* spp.	5	0	0	0
*Alternaria* sp./unknown sp. 5	15	0	0	0
*Aspergillus* sp. section *Nidulantes*	0	0	5	0
*Aspergillus* sp., series *Versicolores*	5	0	10	20
*Beauveria* sp.	5	0	5	0
*Bjerkandera adusta*	5	0	0	0
*Botrytis* spp.	0	0	5	5
*Cephalotrichum* sp.	0	0	10	0
*Cladosporium* spp.	135	45	155	55
*Parengyodontium album*	0	0	55	0
*Mucor plumbeus*	0	0	5	0
*Penicillium* sp., section *Chrysogena*	15	>26,280	0	40
*Penicillium* sp., section *Fasciculata*, series *Camemberti*	0	0	375	0
*Penicillium* sp., section *Penicillium*	0	0	0	395
*Penicillium* sp. 1	5	0	0	0
*Penicillium* spp.	40	0	20	0
*Rhizopus stolonifer*	0	0	0	5
*Scopulariopsis* sp.	0	0	5	0
yeast sp. 1	0	0	5	0
yeast sp. 2	5	0	0	0
yeast spp.	25	0	0	0
unknown sp. 2	0	0	100	0
unknown sp. 3/*Penicillium freii*?	5	0	0	0
non sporulating spp.	10	0	35	15
sum	275	>26,280	790	535

**Table 4 pathogens-13-00022-t004:** The resulting *p*-values of the ANOVA-s used on the radial growth of strains. Significant results are marked with ***: *p*< 0.001, **: *p*< 0.01, and *: *p*< 0.05.

Fungal Taxon	Fungal Strain	Water Activity		pH		Red Mud	
*Penicillium* sp.	T398A	6.91 × 10^−19^	***	1.74 × 10^−12^	***	1.02 × 10^−2^	*
*Aspergillus* sp., series *Versicolores*	T398B	1.55 × 10^−14^	***	1.47 × 10^−13^	***	1.32 × 10^−3^	**
*Penicillium* sp.	T399	5.06 × 10^−18^	***	5.49 × 10^−14^	***	3.88 × 10^−4^	***
*Penicillium chrysogenum*	T401A	2.00 × 10^−15^	***	4.74 × 10^−9^	***	3.74 × 10^−1^	
*Penicillium* sp.	T401B	1.70 × 10^−15^	***	3.13 × 10^−13^	***	1.35 × 10^−1^	
*Aspergillus* sp., series *Versicolores*	TD12	3.73 × 10^−26^	***	1.58 × 10^−14^	***	1.16 × 10^−1^	
*Penicillium* sp., section *Penicillium*	TD16	4.85 × 10^−18^	***	2.68 × 10^−13^	***	2.97 × 10^−57^	***
*Penicillium buchwaldii*	TD22	6.15 × 10^−15^	***	1.27 × 10^−11^	***	3.74 × 10^−1^	
*Mucor circinelloides*	TD25	2.00 × 10^−24^	***	1.18 × 10^−13^	***	1.69 × 10^−2^	*
*Aspergillus* sp. section *Nidulantes*	TD4	2.69 × 10^−26^	***	1.62 × 10^−20^	***	1.00 × 10^0^	
*Beauveria* sp.	TD5	2.16 × 10^−22^	***	3.56 × 10^−16^	***	4.42 × 10^−1^	
*Cephalotrichum* sp.	TD6	3.74 × 10^−30^	***	4.35 × 10^−23^	***	1.16 × 10^−1^	
*Penicillium* sp., section *Chrysogena*	TD8	1.17 × 10^−15^	***	4.74 × 10^−16^	***	9.64 × 10^−4^	***
*Aspergillus* sp., series *Versicolores*	TD9	3.83 × 10^−22^	***	8.71 × 10^−14^	***	1.32 × 10^−2^	*
*Penicillium expansum*	SZMC2725	-	-	-	-	1.52 × 10^−5^	***

**Table 5 pathogens-13-00022-t005:** Chemical composition of the red mud sample used in growth tests.

Elements		Concentration		Wavelength of Detection (nm)	Limit of Detection (mg/kg)
Al	Aluminium	g kg^−1^	31.8	394.401	0.5
Ca	Calcium	g kg^−1^	54.1	317.933	2.5
Fe	Iron	g kg^−1^	129	259.939	0.5
Mg	Magnesium	g kg^−1^	7.9	279.077	5
K	Potassium	g kg^−1^	0.79	766.49	10
Na	Sodium	g kg^−1^	13.7	330.237	5
As	Arsenic	mg kg^−1^	53.3	188.979	1.25
Ba	Barium	mg kg^−1^	48.5	233.527	0.5
Cd	Cadmium	mg kg^−1^	1.23	228.802	0.05
Co	Cobalt	mg kg^−1^	45.8	228.616	0.05
Cr	Chromium	mg kg^−1^	273	267.716	0.05
Cu	Copper	mg kg^−1^	35.0	327.393	0.05
Hg	Mercury	mg kg^−1^	<0.5	194.168	0.5
Mn	Manganese	mg kg^−1^	2376	257.61	0.1
Mo	Molybdenum	mg kg^−1^	< 0.5	202.031	0.5
Ni	Nickel	mg kg^−1^	114	231.604	0.2
P	Phosphorus	mg kg^−1^	772	214.914	5
Pb	Lead	mg kg^−1^	68.4	220.353	0.2
S	Sulfur	mg kg^−1^	1605	181.975	10
Se	Selenium	mg kg^−1^	<5.0	196.026	5
Sr	Strontium	mg kg^−1^	183	407.771	0.05
Zn	Zinc	mg kg^−1^	79.2	213.857	0.05

## Data Availability

The data presented in this study are available on request from the corresponding author.

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
