# Peer review of "Characterization of Indoor Molds after Ajka Red Mud Spill, Hungary"

_pathogens, 2023, doi:10.3390/pathogens13010022_

Round 1

Reviewer 1 Report

Comments and Suggestions for Authors

 This study investigated the effects of red mud on indoor mold growth by  conducting experiments with three case study buildings in Devecser affected by the red mud catastrophe. A total of 43 fungal taxa were detected in the building with varying concerntrations, and  results revealed how these fungal strains were affected by the red mud. The overall experiments were set up reasonably, and results were presented appropriately. The writing is good with a number of mimor language errors detected. However, I have the following questions which need to be addressed.

1. Does the conclusion applies to other buildings in other regions? can the authors discuss this further by comparing previous studies and the current study.

2. The results indicated that a small number of fungal strains were significantly affected by the red mud, so I think the conclsuison "no difference in mold colonization in the buildings flooded with or without red mud" might be far-fetched. I think the authors need to adjust the expression to make it more convincing. 

3. The introduction section is a bit week, more backgroud informaiton is needed.

4. In terms of all Tables, please indicate full names of all abbreviations 

5   Pay attention to small errors, for example, Line 270,  "[35,38])"

Comments on the Quality of English Language

The overall English language looks good, however, a number of small errors were detected. 

Author Response

This study investigated the effects of red mud on indoor mold growth by  conducting experiments with three case study buildings in Devecser affected by the red mud catastrophe. A total of 43 fungal taxa were detected in the building with varying concentrations, and  results revealed how these fungal strains were affected by the red mud. The overall experiments were set up reasonably, and results were presented appropriately. The writing is good with a number of minor language errors detected. However, I have the following questions which need to be addressed.

1.Does the conclusion applies to other buildings in other regions? can the authors discuss this further by comparing previous studies and the current study.

Response: Thank you very much for your review. We added more information about other flooded buildings in other regions. “Based on the analysis of 28 mold-infested buildings, a trend of higher prevalence of indoor Aspergillus and Penicillium spp. was observed in buildings with previous flooding history. In the indoor environment, flooding history was found as the strongest determinant of the prevalence of the above mentioned genera.”

  1. The results indicated that a small number of fungal strains were significantly affected by the red mud, so I think the conclsuison "no difference in mold colonization in the buildings flooded with or without red mud" might be far-fetched. I think the authors need to adjust the expression to make it more convincing.

Response: Thank you for this remark. We agree. The text was corrected: “Our results indicated that only a small number of fungal strains were significantly affected by the red mud on the artificial media, indicating only a slight difference in mold colonization in the buildings flooded with or without red mud.”

3.The introduction section is a bit week, more backgroud informaiton is needed.

Response: Thank you. We added several new sentences to the introduction about the factors affecting indoor molds, especially water activity and pH, as these ones were studied with growth tests.

  1. In terms of all Tables, please indicate full names of all abbreviations.

Response: Thank you. We have checked the tables for abbreviations. In Table 1, there is one abbreviation (SZMC), so we added the explanation in the footer of the table (SZMC: Szeged Microbiology Collection.). In Table 2 T,S,A,b,w,c and o  are explained in the headings of the 2nd and 3rd columns. ITS: Internal transcribed spacer is added as the explanation of the abbreviation. In Table 3, and 5, there are no abbreviations, except sp., spp., CFU (defined in section 2.1 of the ms) and other common ones which usually are not explained in the mycological papers.

5   Pay attention to small errors, for example, Line 270,  "[35,38])"

Response: Thank you, corrected: ") [35,38]"

The overall English language looks good, however, a number of small errors were detected.

Response: Thank you, we checked the language again, and the errors were corrected.

Reviewer 2 Report

Comments and Suggestions for Authors

The manuscript under consideration presents an interesting research on the biodiversity of fungi colonizing buildings after the red mud catastrophe. In addition, the influence of pH, water activity and red mud concentration on the growth of some fungal strains isolated from the flooded buildings were assessed. Despite the novelty of the obtained data is undutiful, , the presentation of the data should be significantly improved.

My major concerns are the description of experimental design and data presentation. Section 2.2 should be carefully re-written to clarify the experimental design: what factors where tested, what levels of the factors were used, what response variable was measured? Maybe, figure or table with experimental design/conditions would be useful. I also recommend dividing this section into several subsections. The presentation of the results in Table 4 is also unclear. The table is named “Results of radial growth tests”, but it looks like the values in the table are not the growth rate of the fungi. Also, these values are not described in the text. As far as I understand, all the growth measurement results are presented in the Supplementary Materials, while only statistical processing of results is included in the manuscript. This makes it difficult to evaluate the results without detailed description of the comparison procedure used in the text. It is even unclear from Table 4, which factors and their levels positively affect the growth rate and which – negatively. It is also unclear from caption for Figure 2 what values are color coded in the heat-map.

In the Discussion section the biodiversity of fungi colonizing the flooded buildings is discussed, while this section lacks the discussion of the growth properties of fungi determined in this study. Table 5 “Chemical composition of the red mud sample used in growth tests” is also poorly discussed neither in the Results nor in the Discussion sections.

Minor remarks:

L67, L69 and further: please, unify the units according to the requirements of the journal.

L129: please, specify, what concentration was used. Is it mass fraction?

L176: “Psocoptera” should be marked with italics

L178 (Table 2): please, divide the species according to their taxonomy (at least at the level of division).

L182: please, change “P. sp.” to “Penicillium sp.”

Figure 2: please, mark the control samples.

L252: I suppose, there is a misspelling: “…Aspergillus niger can increase the pH the substrate by its excreted acidic metabolites…”

Supplementary figures: please, specify the treatment conditions on each figure instead of digit codes.  

Comments on the Quality of English Language

English is fine, just several misspellings detected.

Author Response

Dear Reviewer, Thank you very much for your work. For a better readibility, we upload our response in a Word file, where  questions and responses are highlighted in yellow and green, respectively. The unformatted text is also available below:

Reviewer2

The manuscript under consideration presents an interesting research on the biodiversity of fungi colonizing buildings after the red mud catastrophe. In addition, the influence of pH, water activity and red mud concentration on the growth of some fungal strains isolated from the flooded buildings were assessed. Despite the novelty of the obtained data is undutiful, the presentation of the data should be significantly improved.

My major concerns are the description of experimental design and data presentation. Section 2.2 should be carefully re-written to clarify the experimental design: what factors where tested, what levels of the factors were used, what response variable was measured? Maybe, figure or table with experimental design/conditions would be useful.

Response: Thank you very much for your review. We added the following text to the beginning of the Methods chapter: As a complex effect of the red mud on fungi can be assumed due to its heavy metal content, pH and water activity, we studied these 3 factors separately. The effects of red mud, pH and water activity on the growth of fungal isolates (N=15) were tested on different agar media. Colony diameters (mm) were measured as response variable (this is emphasized in the text now). A figure with the experimental design is shown in the graphical abstract.

I also recommend dividing this section into several subsections.

Response: Thank you for this suggestion. We divided it to subsections 2.2.2-2.2.4.

The presentation of the results in Table 4 is also unclear. The table is named “Results of radial growth tests”, but it looks like the values in the table are not the growth rate of the fungi.

Response: The reviewer pointed out, that the content of Table 4 is clearly not obvious, for this we are grateful. The numbers in Table 4 are the p-values for the ANOVA-s of different treatments on the different strains. The meaning of the original number format originating from the R environment was the following: 6.19E-19 = 6.19×10-19. This number formatting was changed to the latter shown format. Also, the caption of the Table was modified, so that it is clear that the values in the Table show the p-values for the ANOVA-s.

 Also, these values are not described in the text. As far as I understand, all the growth measurement results are presented in the Supplementary Materials, while only statistical processing of results is included in the manuscript. This makes it difficult to evaluate the results without detailed description of the comparison procedure used in the text. It is even unclear from Table 4, which factors and their levels positively affect the growth rate and which – negatively.

Response: The reviewer pointed out that it is unclear from Table 4, which factors and their levels positively affect the growth rate and which – negatively.

The primary statistical test – an ANOVA – by its nature cannot show the direction and magnitude of the effect of a factor (treatment regime in this case), just that whether at least one of the groups had different mean values, when compared to all other groups in the same treatment regime. Basically it tells us if the treatment regime would have an effect under any of its levels. If the growth of the strains under any of the treatments in the given regime was significantly different in any direction, when compared to the other treatments, the result of ANOVA will be significant. Whether these ANOVAs are significant or not are presented in Table 4, using the resulting p-values.

To gain more information about the direction and the magnitude of the effects of treatments, pairwise comparisons were made using Dunnett’s post-hoc tests, in which all treatment levels of a given treatment regime were compared to the control treatment of that regime. The results of this are shown in Figure 2 and in the supplementary material. As the detailed discussion of the effect of each treatment in each treatment regime on each strain would have significantly bloated the Results and Discussion part of the manuscript, we selected and focused only on those strains, to which the effect of the red mud substrate treatment was significant based on the previously conducted ANOVA (strains: T398A, T398B, T399, TD16, TD25, TD8, TD9, SZMC 2725). This choice was made as the primary focus of the MS was the effect of red mud on the growth of different fungal strains. The following parts of the main text were modified to make this clearer:

This post-hoc analysis enabled a detailed comparison of each treatment group against a control group, pinpointing any treatment exhibiting statistically significant differences in colony diameter. In the case of the red mud agar treatment, where only two levels of treatment were compared, the results of the ANOVA and Dunnett’s test are equivalent, as only one pairwise comparison of two groups was necessary.”

The radial growth test showed statistically significant differences in colony diameters under the different treatment regimes (Table 4, Figure 2, Supplement Figures S1-S43). Salt concentrations (water activity) and pH had significant effect on all strains based on the results of the ANOVA-s (Table 4). Based on the pairwise comparisons to the control treatments done with Dunnett’s tests, two strains (TD8, T401B) tolerated low water activity (aw 0.922), but most of the tested fungi showed poor development on it. Some strains (TD4, TD6, TD9, TD12, TD22, TD25) did not tolerate even moderate aw values (0.954-0.962). High aw was preferred by TD4, TD5, TD6, TD9, TD12, TD22, TD25, and T399 (Figure 2, see Supplementary Material for detailed results of the Dunnett’s test).”

“Most strains preferred the pH range of 5.0-7.0. Low pH was not tolerated by TD6 (pH≤4.0), as well as T398B, TD8 and TD25 (pH=2.2), while the other stains tolerated it poorly, showing delayed and reduced growth. Growth of TD4 was inhibited by pH 8 and 7, while this fungus preferred pH 3. TD5 grew well on all pH values (Figure 2, see Supplementary Material for detailed results of the Dunnett’s test).”

All tested strains were able to grow on red mud agar, including controls. High red mud concentration (10-1 dilution of the original sample) significantly affected the growth of 8 strains (Table 4). The combined effect of different culture media to test red mud, its pH and aw on fungal growth is shown on Fig 2. A weak positive effect of red mud was observed in one case (TD25). Negative effect of red mud was found in 7 strains. Delayed initial growth was detected in 3 control strains (T399, SZMC 2725 and T398A). Smaller colony diameter was measured at the end of the incubation period in TD8, TD9 and TD16, and in the control strain T398B. However, in two of them the negative effect was possibly due to the pH of the red mud media, but not its heavy metal content (see Fig 2, TD9 and TD16). High red mud concentration was tolerated by the remaining strains without any reduction in growth. Based on the growth profile of the individual strains, the 10-4 dilution of red mud had no significant effect on any strains (Figure 2, see Supplementary Material for detailed results).”

It is also unclear from caption for Figure 2 what values are color coded in the heat-map.

Response: Thank you. We rewrote the caption of Figure 2. “The effect of red mud, pH and water activity (aw) on the colony diameters of fungal strains on special media (MEA+10-1 red mud concentration, MEA+McIlvain buffer solutions, and YEA+NaCl, respectively), when compared to their growth on the control media (red mud 10-1 control= MEA+red mud 10-4, pH control=MEA, aw control=YEA,). Ranges of aw (0.97±0.02) and pH (6.50±0.30) corresponding to MEA with 10-1 red mud concentration are selected. Strains, where red mud media had no significant effect are excluded. Significant, positive effects (i.e., growth is more intensive on the special media than on control media) are marked with red, orange and yellow; negative effects are marked with blue colour. ns=non significant; nd=no data (not studied). Significant results are marked with ***: p< 0.001, **: p< 0.01, *: p< 0.05.

In the Discussion section the biodiversity of fungi colonizing the flooded buildings is discussed, while this section lacks the discussion of the growth properties of fungi determined in this study.

Response: Thank you. The following text was added to the discussion: According to our growth tests, most (10 of 15) fungal strains isolated from building materials covered with red mud tolerated its heavy metal content and grew well on the typical range of pH and water activity characterizing red mud substrate. Isolates tolerating red mud poorly belonged to Aspergillus sp. series Versicolores (T398B), Penicillium expansum (SZMC 2725), Penicillium section Chrysogena (T08) and Penicillium spp. (T398A, T 399). Interestingly, isolates well tolerating red mud were their close relatives, e.g., TD12 (Aspergillus sp. series Versicolores) and TD401A (P. chrysogenum). This suggests that red mud tolerance may be determined by infraspecific genetic factors.

Table 5 “Chemical composition of the red mud sample used in growth tests” is also poorly discussed neither in the Results nor in the Discussion sections.

Response: The descriptions of the chemical composition in the Results and in the Discussion sections were expanded. Results: “The results were given in Table 5. The main chemical elements of the red mud sample are Fe, Ca, Al, Na. Converted to oxide, we obtained the following concentrations for the main elements: 37% Fe2O3, 12% Al2O3, 7,6% CaO, 3,8% Na2O. Elements, such as K, Mn, S, P, Cr, Sr, Ni, Pb, Zn, Cu, Co, Cd were found in minor or trace amounts. The concentrations of Mo, Se and Hg were under their respective detection limits.”

Discussion: “The chemical composition of the red mud from Ajka was measured by several studies. The results of the sample in the present study are in good agreement with previous findings (citations).”

Minor remarks:

L67, L69 and further: please, unify the units according to the requirements of the journal.

Response: Thank you, corrected.

L129: please, specify, what concentration was used. Is it mass fraction?

Response: Thank you for pointing out this missing information. The concentration of red mud in MEA was assessed in terms of mass fraction, and this information has been added to the manuscript. „Red mud was added to MEA in different concentrations at 10-1, 10-2, 10-3, and 10-4 m/m%, having pH of 7.0, 6.8, 5.2 and 5.2, respectively. Based on the previous growth tests, MEA media with 10-1 and 10-4 red mud concentrations were selected for further studies.”

L176: “Psocoptera” should be marked with italics

Response: It is an Order, and not italicized.

L178 (Table 2): please, divide the species according to their taxonomy (at least at the level of division).

Response: Thank you – we added divisions to Table 2.

L182: please, change “P. sp.” to “Penicillium sp.”

Response: Thank you, corrected.

Figure 2: please, mark the control samples.

Response: Figure 2 shows no individual data, but relative results compared to the control samples. Control results are shown in Supplementary Materials. However, the caption of the Figure 2. is rewritten in order to avoid misunderstandings.

L252: I suppose, there is a misspelling: “…Aspergillus niger can increase the pH the substrate by its excreted acidic metabolites…”

Response: Thank you, corrected. Aspergillus niger can decrease the pH of the substrate by its excreted acidic metabolites

Supplementary figures: please, specify the treatment conditions on each figure instead of digit codes.

Response: Thank you. Instead of the digit codes, the following text was added to each figure in the Supplementary material: aw =0.997, aw =0.991, aw =0.980, aw =0.968, aw =0.962, aw =0.951, aw =0.945, aw =0.922, red mud 10−1, red mud 10−4, pH = 2, pH = 3, pH = 4, pH = 5, pH = 6, pH = 7, pH = 8.

English is fine, just several misspellings detected.

Response: Thank you, we checked the language again, and the errors were corrected.

Round 2

Reviewer 2 Report

Comments and Suggestions for Authors

The manuscript was significantly improved. I have just two minor remarks:

L164-165: please, check the units of the concentration. I suppose, it should be M not mol?

I also highly recommend to add some more information in the Conclusions section to highlight the significance of your research.

Comments on the Quality of English Language

English is fine, just needs a spell check.

Author Response

Reviewer: The manuscript was significantly improved. I have just two minor remarks: L164-165: please, check the units of the concentration. I suppose, it should be M not mol?

Response: Thank you for your suggestion. It is corrected. “Mol” was replaced with “M”: “mixtures of 0.3 M Na2HPO4 × H2O and 0.1 M citric acid in different proportions”

Reviewer: I also highly recommend to add some more information in the Conclusions section to highlight the significance of your research.

Response: Thank you, we agree, the conclusion should be more informative. The following text was added to the conclusion: “The heavy metal content of red mud had a negligible role in the inhibition on fungal growth or composition, but other aspects, e.g. modification of metabolite production is still largely unknown. Among the many negative outcomes of floodings combined to chemical pollution, the effects of fungal colonization may be complex and hardly predictable, therefore, extra caution should be used.”